# Ecological Impacts Associated with the Qinghai–Tibet Railway and Its Influencing Factors: A Comparison Study on Diversified Research Units

**DOI:** 10.3390/ijerph20054154

**Published:** 2023-02-25

**Authors:** Lili Zhang, Yi Miao, Haoxuan Wei, Teqi Dai

**Affiliations:** 1Faculty of Geographical Sciences, Beijing Normal University, Beijing 100875, China; 2College of Geography and Environment, Shandong Normal University, Jinan 250061, China

**Keywords:** ecological impacts, influencing factors, comparison, Qinghai–Tibet Railway

## Abstract

The ecological impacts of the construction and operation of the main transport infrastructure on the Qinghai–Tibet Plateau cannot be disregarded. Based on different sections, buffers, bilateral sides, and periods, the authors of this study explored the ecological changes along the Qinghai–Tibet Railway through an integrated analysis of the landscape fragmentation index and ecological service value calculation from 2000 to 2020, as well as the influencing factors of differentiated trends, using multinomial logistic regression. It was discovered that there was heterogeneity among the sections, buffers, and bilateral sides in both the landscape fragmentation index and the ecological service value. It was also found that there was recoverability in the operation period, compared to the construction period. The negative correlation between the landscape fragmentation index and the ecological service value was only significant in 2020, which was not enough to fully explain the negative effect between them. Distinct human and natural circumstances have resulted in different consequences. However, regions far away from the main settlement areas, and with lower population densities, could aid in the simultaneous recovery of the ecological service value and landscape fragmentation index. According to these findings, prior studies may have exaggerated the ecological impact of the Qinghai–Tibet Railway. However, it should be highlighted that, in a location with a delicate ecological environment, it is still crucial to consider regional development, infrastructure construction, and ecological protection synchronously.

## 1. Introduction

The public—and academics—have grown increasingly concerned with ecological changes as a result of the rational understanding of sustainable development. The construction and development of transport infrastructure, which serves as the foundation and direction of regional development, have significant effects on the regional ecosystem. As a result, the area of transport ecology has evolved [1,2]. Early relevant studies have mostly been conducted in European and American nations [3,4]. It was determined that transport networks, connectivity, and trunk lines have ecological impacts, including disruptions to the ecosystem, contamination of the environment, reduced vegetation cover, risk to human health, interference or cutting of the surrounding landscape, habitat degradation, and even changes to the structure of the ecosystem and loss of potential ecological function [5,6,7,8,9]. Additionally, some research has shown that transport disrupts horizontal ecological flow, which changes the spatial pattern of animal behavior and even reduces biodiversity [10,11,12,13].

China has recently become the focus of significant attention due to the rapid construction of its transportation system. Studies on the effects of transport infrastructure on ecosystems in China have often focused on Central and Eastern China, and the majority of them have concurred on the negative effects of transportation on ecology [14,15]. Through empirical research, it was determined that ecological landscape fragmentation was highly related to the road grade and transport network and its centrality and density [16,17,18,19]. The degree of landscape fragmentation dramatically increases with an increase in road density. Additionally, there have been notable differences in the effects of transport construction and operation on the biological landscape [20,21]. However, the pertinent research needs to be further enriched regarding large-scale fragile ecological environment areas.

This paper focused on evaluating ecological changes along the main transport lines on the Qinghai–Tibet Plateau (QTP). The QTP, also known as “the roof of the world”, is the highest plateau in the world and the headstream of the majority of Asia’s rivers. Given the regional (and perhaps worldwide) ecological importance of this area, the construction and development of transportation systems must consider ecological protection. Meanwhile, transport construction on the QTP has been restricted because of its complicated and notable regional features, such as its remote location, widespread permafrost, and sparse population [22,23,24]. However, relevant studies discovered that the region’s transport infrastructure directly changed the land use pattern, disrupted ecosystems, increased the ecological vulnerability and landscape fragmentation, and impaired the function and stability of local ecosystems, even though different transportation modes had different ecological impacts [25,26,27,28]. Researchers discovered impacts of the Qinghai–Tibet Railway on ecological disturbance, terrain patterns, environmental pollution, and wildlife [26]. Additionally, variance among different locations, such as different sections or buffers along the transport lines, were discovered. For instance, recent research suggested that the effects of trunk lines on landscape fragmentation on the QTP were concentrated within a 15 km buffer, whereas the impact on vegetation abundance was only present within the first 5 km. Additionally, the research suggested the latter impact was more pronounced during the construction phase, and that losses in primary productivity and biomass within the first kilometer of buffer were much higher [11,25,29,30].

Generally speaking, pertinent studies have primarily utilized multiple data sources, including remote sensing images such as Landsat and GIS technology, as well as multivariate ecological indices, such as the patch index, value function, and spatial indicator function or field sampling, for analysis [31,32,33,34]. However, there have been insufficient empirical investigations exploring the connections between the landscape ecological indicators and the ecological service value indicators, particularly the direct influence of the construction period and the prospective long-term impact of the operating period. In addition, previous research has tended to evaluate different buffers rather than different sections and sides of the transport line. Additionally, land use data, with 1 km-per-pixel precision, were frequently employed in earlier research on the ecological impacts of large-scale transport infrastructure, including the Qinghai–Tibet Railway and highway. However, more precise data could be adopted with the development of remote sensing technologies.

As an ecological functional area with a fragile ecological environment, it is worth discussing whether the construction and operation of large facilities on the Qinghai–Tibet Plateau would reduce its ecological service value or aggravate its fragmentation. The primary objective of this study was to thoroughly analyze the ecological impacts of different buffers, sections, and both sides during the construction and operation periods of the Qinghai–Tibet Railway, as well as the spatiotemporal differentiation characteristics. This analysis was based on transport vector data and land use data with 30 m precision. This study also used multinomial logistic regression to further assess the influence of natural, socioeconomic, and locational factors on the ecological changes in the different study units [17,30]. Based on the availability of data, we selected three research periods, based on the land use data of Globeland30 (v2000, v2010, and v2020) and the construction process of the Qinghai–Tibet Railway. The v2000 data were from the base year of 2000, which corresponded to the relative initial stage of the Qinghai–Tibet Railway. At that time, the only constructed section of the Qinghai–Tibet Railway was between Xining and Golmud, while that of Golmud to Lhasa had yet not been constructed. The v2010 data were from the base year of 2010 and were produced from 2008 to 2011. In terms of the possible impact of the Qinghai–Tibet Railway, the main differences between the v2000 and v2010 data were due to the completion of the whole line from 2001 to 2006 and the initial opening of the section between Golmud and Lhasa in 2006. The v2020 data were produced from 2017 to 2020. In terms of the possible impact of the Qinghai–Tibet Railway, the differences between the v2010 and v2020 data were due to the operation of the entire Qinghai–Tibet Railway during this period.

This paper was separated into four sections. The study area, data, and methods were described in Section 2. In Section 3, we analyzed the landscape fragmentation and ecological service value in various sections and buffers of the Qinghai–Tibet Railway, as well as the correlation relationship and the influencing factors. Section 4 was devoted to a discussion of our findings from various angles, and the conclusion of this paper as presented in the Section 5.

## 2. Materials and Methods

### 2.1. Study Area and QTR

The Qinghai–Tibet Plateau (QTP), also referred to as “the roof of the world” and “the third pole of the Earth”, is situated between 25°59′37″ N and 39°49′33″ N and 73°29′56″ E and 104°40′20″ E (Figure 1), with an elevation range of 1130 to 6894 m and a surface area of approximately 2,542,400 km^2^. It should be underlined that the QTP is a very significant, ecologically sensitive area, and an ecologically functional one. For instance, its geology is young and active, making it susceptible to geological disasters. In addition, the QTP contains the headwaters of the majority of Asia’s rivers, and therefore, macroregional sustainable development is greatly impacted by its ecological conservation. The QTP’s high altitude, chilly climate, and widespread permafrost have had a tremendous impact on a variety of human activities and have severely hindered the growth and development of transport infrastructure.

The Qinghai–Tibet Railway (QTR) was the first railroad to connect Tibet’s hinterland and is one of the most crucial regional, large-scale transportation arteries on the QTP. In 2006, its full operation was achieved, and from 2007 to 2011, the double-track construction from Xining to Golmud was completed. This railway connects Lhasa, the capital of the Tibet Autonomous Region, to Xining, the capital of Qinghai Province. It has a total length of 1956 km and was built in two stages. The first part of the QTR extends 814 km, ranging from Xining to Golmud. Construction started in 1958 and was completed in 1984. Its double-track construction started and was completed between 2007 and 2011. The second part is 1142 km long; it was started in 2001 and finished in 2006, running from Golmud to Lhasa.

The ecological implications of the QTR are a significant topic due to the uniqueness, sensitivity, and susceptibility of the ecological ecosystem there, as well as its length and breadth. According to the research of Zhang et al. (2002) and Miao et al. (2021), we split the QTR into 32 sections, starting from Lhasa to Xining, in this study, in order to examine the ecological impacts and changes along the QTR. Each railway section was 61.1 km long, on average. Additionally, we established 22 buffer widths, as shown in Figure 1, to create a study area with a total size of 182,348 km^2^.

### 2.2. Data Collection

In this study, the vector data of the QTP were obtained from TPDC (data.tpdc.ac.cn (accessed on 1 December 2022)), and the vector data of the QTR were obtained from OpenStreetMap (download.geofabrik.de/asia/china.html (accessed on 1 December 2022)) and compared with BaiduMaps (map.baidu.com (accessed on 1 December 2022)). The data of the land use classifications, with a pixel size of 30 × 30 m, were divided into 10 categories (farmland; forest; grassland; shrubland; wetland; water body; tundra, desert and grassy marshland; artificial surface; bare land; glaciers and perpetual snow) and obtained from Globeland30 (www.globallandcover.com (accessed on 1 December 2022)). Globeland30 involved 3 versions of datasets, including V2v2000 (taking 2000 as the base year), V2v2010 (taking 2010 as the base year and mainly based on land use data from 2009–2011), and V2v2020 (produced since starting in 2017 and published in 2020).

Terrain factors, such as the altitude and slope, were based on digital elevation model (DEM) data (90 m × 90 m, SRTM3), which were obtained from the resource and environment data center of the Chinese Academy of Sciences (www.resdc.cn (accessed on 1 December 2022)). Among them, the value of the DEM was directly used for altitude, and the slope was calculated based on the DEM data through using the slope tool in ArcGIS. The National Tibetan Plateau/Third Pole Environment Data Center and the Resource and Environment Data Cloud Platform of the Chinese Academy of Sciences provided annual temperature and precipitation datasets with a resolution of 1 km [35,36,37]. Spatial Euclidean distances were used to calculate the distances to provincial capitals, prefecture-level cities, and counties. The LandScan dataset (https://landscan.ornl.gov (accessed on 1 December 2022)) was used for the population density. Nighttime light data, including the annual stable DMSP/OLS (2000, 2010) and NPP/VIRS (2020) data, which were processed to a resolution of 0.1 km, are available from NOAA (http://ngdc.noaa.gov/eog/download.html (accessed on 1 December 2022)).

### 2.3. Analysis Framework and Methods

The purpose of this study was to determine the ecological impacts of the QTR and analyze its influencing factors (Figure 2). The authors of this study first obtained the QTR and land use classification data and then employed the landscape index and ecological service value accounting methods, computing the outcomes of the landscape fragmentation index and the average ecological service value of each study unit along the QTR and performing a spatiotemporal analysis. Second, multinomial logistic regression analysis was used to determine the contributions of various natural and human factors on ecological changes along the QTR in different periods. Additionally, this study was separated into the following two periods based on the data’s accessibility and the development of the QTR: the period of the QTR’s construction, which ran from 2000 to 2010, and the period of operation between 2010 and 2020.

In this study, the landscape fragmentation index and ecological service value were used to judge ecological changes. Numerous previously-conducted studies demonstrated that, to a certain extent, the degree of landscape fragmentation reflects the degree of human disturbance to the landscape, whereas the ecological service value refers to the advantages that humans derive directly or indirectly from the ecosystem and serves as the foundation for both ecological protection and ecological function zoning. Relevant studies primarily used the landscape fragmentation index to analyze and judge the cuts or more profound damage of traffic facilities to the landscape pattern, but the indicator results mainly showed the quantitative results, based on changes in land patches, lacking the integrated judgment of the nature and function of different land use types. Therefore, further combining the results of the ecological service value indicators could more comprehensively, and more accurately, reveal the changes and effects caused by the construction and operation of facilities.

First, the term landscape fragmentation, used herein, denotes the transformation of once-continuous landscape elements into several discontinuous areas. This can occur naturally or as a result of human activity disruptions. Following previous studies, the landscape fragmentation index (*LF*), a widely used traditional indicator in landscape ecology, was employed. This is the ratio of the number of patches *n_i_* to patch area *A*, which indicates the segmentation degree of patches in a landscape system. Larger values correspond to more patches per unit area, and thus, a greater degree of fragmentation. The formula is as follows [30]:(1)$$LF=\overset{m}{\underset{i=1}{\sum }}\frac{n_{i}}{A_{i}}$$
where *i* stands for different land use types, *n_i_* and *A_i_* represent the amount and the area of landscape patches of land use type *i* in each study unit, respectively.

Secondly, the ecosystem service value represents the function of an ecosystem. Its calculation was based on the basic equivalent table of ecosystem service function value per unit area, as shown in Table 1, which was consistent with the existing research on the QTP [38]. Thus, the average ecological service value (*AEV*) of each study unit was calculated by the different types of land use and the corresponding ecological service function value per unit area. The calculation formula was as follows:(2)$$AEV=\frac{EV}{A}=\frac{1}{A}\underset{i}{\sum }A_{i}\cdot Va_{i}$$
where *AEV* indicates the average ecological service value, and its measurement unit is CNY(￥)/hm^2^. *A_i_* is the area of land use type *i*, while *Va_i_* is the ecological service value per unit area of land use type *i*. The specific ecological value assignment was based on the existing research results, as shown in Table 2.

Based on the aforementioned classification, we divided the ecological changes along the QTR into four categories, based on the expansion or contraction of the *AEV* and *LF* throughout two study periods: 2000 to 2010 and 2010 to 2020. Then, we utilized a multinomial logistic regression model with the aforementioned four categories as dependent variables to analyze and interpret the data. A logistic regression model is a practical approach that works well for classifying dependent variables, and it is frequently employed in related investigations [39,40]. Some researchers included spatial variables as well, to learn more about the effects of location and spatial relationships [41]. Multinomial logistic regression was used in this research to further investigate the affecting factors, as shown in Formula (3):(3)$$ln\left[\frac{P_{n}}{P_{0}}\right]=\alpha_{n}+\sum_{k=1}^{14}\left(\beta_{nk}\cdot x_{nk}\right)\;,n=1,\;2,\;3$$
where *P*_0_, *P*_1_, *P*_2_, and *P*_3_ are the probabilities of occurrence of the interpreted variables *Y*_0_, *Y*_1_, *Y*_2_, and *Y*_3_, respectively. Among the interpreted variables, *Y*_0_ was the control group. Correspondingly, *P*_0_ was used as the reference for logit transformation. In addition, *k* represents a type of independent variable, including 14 categories. *α* is a constant, and *β* is the fitting coefficient of each independent variable. Here, if *β* shows a positive value, then e*^β^* > 1, indicating that it can result in an increase in *P* and the occurrence of *Y*. In this study, 14 independent variables were selected. These are shown in Table 2, which is based on the nonnegligible and distinctive environmental conditions of the QTP, as well as existing relevant research [16,17,18,19,30].

## 3. Results

### 3.1. Landscape Fragmentation in Different Units

We discovered an inverse relationship between the amount of landscape fragmentation and the breadth of the buffers to an extent, forming a power-law-like distribution (Figure 3). In general, the *LF* decreased as the width of the buffers increased. Buffers with widths larger than 2 km tended to be stable. The above results suggested that the consequences of the QTR on the landscape might be restricted in areas closer to the railway, which was similar to the findings of the existing research.

Differences between the buffers on the bilateral sides were also noticeable. Specifically, the buffers on the right side had a relatively low *LF* and continuously decreased, except for the 0.1 km buffer. In contrast, the left side’s buffers first went through fragmented increases from 2000 to 2010, then generally decreased in 2020, and then performed even better than in 2000. These results indicated that the construction of the QTR may have had more obvious landscape impacts on its left side’s buffers, while the ecological landscape pattern was gradually restored in the operation period from 2010 to 2020 for most buffers.

Additionally, the disparities among the 32 sections were also obvious, as were those between both sides of the railway (Figure 4). Sections 1~3, left side buffers of sections 4~13, and those around section 26 showed a more fragmented landscape pattern. Meanwhile, the results of the *LF* of the right side buffers in sections 4~6 and of both sides’ buffers in sections 16~19 and 29~32 were relatively minor. However, the results of the three years were similar, on the whole.

Figure 5 depicts the spatial distribution of landscape fragmentation. As a further detailed display of the above results, the distribution pattern was consistent with the outcomes of different road sections and buffers. First, it is worth noting that the overall distribution pattern of the *LF* was not obviously altered. Additionally, since the QTR did not begin construction in the southern sections, the results in 2000 demonstrated that the terrain in this region became significantly fragmented. This further suggested that the QTR’s effects on the ecological landscape may have been limited. Furthermore, throughout this time, the *LF* decreased in the majority of study units, and the ecological landscape patterns along the railway could be restored in the operation stage.

Secondly, different buffers along the same road sections could be remarkably consistent or quite distinctive. For instance, in the third, twenty-fifth, and twenty-seventh sections, the results of the study units that were close to the railway were obviously different from those far away from the railway, whereas there were almost no differences among the different buffers or sections in the middle of the QTR or those around Xining.

Contrary to expectations, there did not appear to be a sufficient trend of a more seriously fragmented landscape closer to the railway. The increase in the *LF* was also not concentrated around the QTR. This finding indicated that the *LF* along the QTR was not primarily caused by the railway, either in the construction or operation periods.

### 3.2. Ecological Service Value in Different Units

Figure 6 displays the average ecological service value (*AEV*) for different buffers along the QTR, indicating that, to a certain extent, there was a positive association between the *AEV* and the buffer width, although the difference between the two sides of the railway was very obvious.

For most study units, there was a clear change in the *AEV* findings over time, which declined from 2000 to 2010 and increased from 2010 to 2020. However, the relative disparities between the different buffers were generally consistent. The *AEV* was obviously the lowest for the 0.1 km buffers. When the buffer width exceeded 0.5 km, the *AEV* results then had an upward trend.

However, the buffers on the left and right sides differed significantly from one another. When the buffers on the left side expanded to a width of less than 13 km, the results of the *AEV* were rather stable and even decreased, but when the buffers were wider than 13 km, they increased dramatically. The *AEV* data showed an upward trend while the right side’s buffers expanded, but they also fluctuated and started to decline after 15 km. Additionally, the increases were most noticeable around the 12 km buffer on the left and the 10–40 km buffer on the right.

Disparities were also evident among 32 sections and on both sides of the QTR (Figure 7). Specifically, the right sides of several northern parts had advantages over other sections. The left side of the third section also had a higher *AEV*. As a result, the *AEV* clearly differed in each section. From the perspective of the correspondence of spatial distribution, there were also great differences between the *AEV* and *LF*.

The three study years’ distribution patterns were rather stable, as seen in the spatial distribution of the *AEV* in Figure 8. It was obvious that, in this spatial distribution, the extremely impressive *AEV* values for the right-side buffers along sections 30–32 were closely related to the position of Qinghai Lake, the largest lake in China.

It is also important to note that, from 2000 to 2010, the *AEV* along the QTR typically decreased, and sections 7–18 and 22–32 of the QTR had *AEV* scores that more obviously decreased closer to the railway. In some ways, this demonstrated that some areas’ *AEVs* were more sensitive to the QTR construction. However, the overall results indicated dramatic improvement from 2010 to 2020, suggesting that it could be possible to restore the environment along the railway during its operational time.

### 3.3. Correlation between LF and AEV

To further clarify the relationship between the *LF* and *AEV*, the authors of this paper carried out a Pearson correlation test, and the results are shown in Table 3. From this, it was inferred that the Pearson connection between the *AEV* and *LF*, along with their growth over the course of the two study periods, was relatively limited. Only 2020 exhibited a discernible negative association, and its *p*-value was in the range of 0.01–0.05. This suggested a strong negative correlation between them. However, it may not have been sufficient to fully define the negative correlation between the *LF* and *AEV* or to explain why the intensification or mitigation of the *LF* led to a reduction or increase in the *AEV*. For instance, section 3’s buffer on the left had high *LF* and *AEV* scores, while section 20 exhibited poor *LF* and *AEV* scores. In general, the relationship between the *AEV* and LF results was particularly complicated, and it was necessary to thoroughly analyze how the *AEV* and *LF* evolved in different study units by incorporating more components.

### 3.4. Influencing Factors Analysis

Multinomial logistic regression analysis was used to identify the explanatory power or contribution effect on ecological changes along the QTR in related natural and socio-economic factors and on the location. The goodness of fit (R^2^) values of the results for the time periods of 2000 to 2010 and 2010 to 2020 were 37.7% and 36.0%, respectively, which showed that explanatory variables had a specific interpretation influence on the dependent variables. Additionally, all selected variables passed the collinearity test, with all *VIF* values lower than 10. Table 4 displays the results of the fitting. 

The results from the QTR’s construction period were as follows. According to the topographic factors, the altitude had a substantial negative impact on the occurrence of Y1 and a significant positive impact on Y2, showing that Y1 was more likely to occur in regions with a lower altitude. The slope had a significant positive effect on Y2 and Y3, showing that, during the construction period, the increase in the *AEV* was more noticeable in areas with high topographical relief. It could be said that, during this time, the smaller the slope in the area, the more noticeable the decline in *AEV*, perhaps because human activity was distributed more densely in places with moderate terrain. In terms of climate, Y1 was more likely to appear when there was more precipitation, but it had a significant negative effect on Y3. Temperature, however, significantly benefited Y3 during this period. The QTP’s chilly climate and widespread permafrost had a tremendous impact on a variety of human activities and severely affected ecological and environmental changes during the construction of QTR. Non-permafrost areas showed a significant positive effect on Y2. From the perspective of location conditions, Y1 was more likely to happen close to provincial capitals, Y2 was more likely to happen close to counties, and Y3 was more likely to occur far away from the provincial capitals but near the counties. Additionally, locations with robust economic vitality had a higher likelihood of forming Y2. This indicated that, during the construction of the QTR, although the linear cutting of the railway project on the ecosystem had an impact on the ecosystem, it was very sensitive to human disturbance. Additionally, it was shown that the sequence of the railway sections had a substantial and significant positive impact on the formation of Y3, indicating that the northern sections were more likely to have good ecological landscape evolution. 

During the operation period, the negative effects of the altitude on the appearance of all types of dependent variables, except the control group, were significantly enhanced, with a negative fitting coefficient of the altitude, possibly indicating that the control group was more likely to occur in areas with higher altitude. Although the slope had a positive impact on Y1 and Y2, it tended to have a negative impact on Y3, suggesting that Y3 might have occurred concurrently in areas with significant or minor surface undulations during this period. Overall, the adverse impact of terrain resistance on the *AEV* increased. More precipitation contributed to the emergence of Y3, while the disaster vulnerability had a significant inhibitory effect on the formation of Y2. It was observed that the high altitude, low rainfall, and vulnerability of the QTP would not be conducive to the restoration of the ecological environment, and manual intervention after construction would be particularly important for the restoration of the ecological environment of the QTR. In terms of location, Y1 was most likely to take place away from prefecture-level cities but close to the province capitals. Y2 was most likely to occur in regions removed from prefecture-level cities and counties. Y3 was likely to occur in areas far away from both provincial capitals and prefecture-level regions and counties. On this basis, it was advantageous for the formation of Y2 and Y3 to be remote from counties and prefecture-level cities. From the standpoint of population and regional socio-economic development, strong economic vitality prevented the formation of Y1 and Y2, while a huge population prevented the emergence of Y3. It was observed that human activities had a great impact on the ecological environment restoration of QTR. In addition, the southern sections were more likely to contribute to the appearance of types except for the control group. However, the width of the buffers and the sides of the QTR explained the dependent variables, but had no appreciable positive or negative impacts. Wholly speaking, the permafrost regions and non-permafrost regions had weak explanatory power for the *LF* and *AEV*. The explanatory powers of different sections were greater than those of different buffers and sides, which may be attributable to the significant differences in ecological environment in different sections of the QTR. Construction and regional development may have aggravated *LF* and *AEV*, on the whole.

## 4. Discussion

This study’s objective was to explore ecological impacts along the QTR and their temporal and spatial aspects from the perspective of transport geography and a systematic analysis of the human–land relationship. We refined the study units by separating the 22 buffers, 32 sections, and the left and right sides of the QTR in this study, and made extensive comparisons by separating the construction and operation periods. In comparison to earlier research, we felt it was crucial to discuss the effects of transportation on ecology through a diverse and refined division of study units, which served as the foundation for obtaining more precise and in-depth data with which to evaluate the ecological impacts of the QTR.

The findings indicated that the *LF* and *AEV* along the QTR varied between different segments and buffers, and their effects may have only been observed in close proximity to the railway. The aforementioned results were consistent with earlier investigations [25,29,30]. However, in contrast to the studies mentioned above, we discovered that the influence of the QTR on ecological impacts may have been less severe. The results of the distinct periods were also different from what was expected. There did not appear to be enough proof that the closer to the railway, the more serious the trend of ecological damage [11,27]. The overall distribution pattern of the *LF* and *AEV* did not significantly change in this study. This finding suggested that the ecological impacts along the QTR were not primarily due to the railway, either during its construction or operation. Even during its construction, the QTR’s influence on the surrounding environment may have been minimal.

It has long been assumed that there a negative link must exist between the value of ecosystem services and the amount of landscape fragmentation. This has been observed in earlier studies that have indicated that landscape fragmentation was accompanied by a reduction in the ecological service value [42,43]. However, this conclusion was not entirely supported by the findings of this study. Although the results in 2020 showed a negative correlation tendency, there was not enough evidence, in this study, to determine whether the improvement or mitigation of the *LF* resulted in a decrease or an increase in the *AEV*. Most research units demonstrated a clear improvement and restored trend, particularly throughout the operational period. Especially compared to highways [30], its ecological impact has proven more recoverable. With the progress in technology, it could also be possible to further control the ecological impacts, and strengthening the railway connection could be eco-friendlier. The lengths of the railway sections and the range of the buffers may have led to changes in the results, but these changes should not fundamentally reverse the above results. This demonstrated that previous research may have overstated the negative effects of regionally extensive transport infrastructure development on the environment. Although the QTR’s construction and operation have directly altered how the land is used, our findings demonstrated the possibility that neither would have a significant long-term or spatially extensive impact on the ecosystem. These results were consistent with the optimistic predictions made by Shen (2004) and Wang (2015) based on observed decreases in ecological impacts [29,44]. Of course, there could be differences in the locations or modes of transportation [30,45,46]. We also discovered that, similar to previous studies, the effects of the QTR on ecosystems were more pronounced in locations close to major towns with larger population densities [18,47,48]. Additionally, effects of the terrain and climate were shown. However, the fitting effect was not particularly ideal and could have been affected by unstable climate change and meteorological changes.

In road network construction in European and American countries, more attention is paid to animals and plants, landscape ecology, resource utilization, etc., while domestic research on road network construction and ecological environment design is relatively simple [49]. Although the linear cutting of the ecosystem by railway engineering will affect the ecosystem, the vulnerability of the alpine ecosystem on the QTP makes it very sensitive to human disturbance. Additionally, its recovery speed is relatively fast after the end of the engineering activities. On this basis, combined with the road network model of Forman and ecological environmental protection measures in other areas, we believe that, during the construction period of the QTR, strict vegetation protection measures should be taken [29], with reasonable planning and construction of walkways and sites, to prevent excessive damage to the construction sites and construction camps. In the operation period, continuous monitoring and effect evaluation should be carried out on the QTR ecosystem, and appropriate manual intervention should be taken to promote the restoration of the plateau ecosystem. Desert and bare-land areas with less human activity present challenges to ecological restoration efforts, and should be a focus. In addition, railway route selection design should avoid ecologically fragile and difficult-to-restore areas. At the same time, compared to other areas, artificial intervention after construction could be particularly important for ecological environmental restoration of the QTR.

In conclusion, the QTP is a typical ecologically fragile and underdeveloped location. As such, it deserves special consideration in the study of the complex effects of transportation on delicate eco-environmental systems. The findings of this study could provide some reference for research on the relationship between landscape fragmentation and the value of ecosystem services. The results could also have an impact on ecosystem-related policies in some regions and provide some reference for the development of transport infrastructure. There are numerous, ongoing, large-scale transport and infrastructure projects being built on the QTP as part of China’s goal of high-quality regional development and the wellbeing of its citizens. As we determined that the ecological impact range of QTR was limited and recoverable, this study could prove beneficial to efforts to improve the precise ecological restoration control of the Sichuan–Tibet Railway, and could serve in the construction and promotion of new lines, such as the Xinjiang–Tibet Railway. The ecological effects covered in this paper could serve as a useful source of motivation for actively promoting construction of the Sichuan–Tibet Railway and for more infrastructure construction for people’s livelihood and welfare. They could also be used as a guide for accurately anticipating potential adverse ecological effects brought on by related transport infrastructure, thus fostering ecological restoration.

It should be noted that, despite the possibility that earlier research has overstated the impact of trunk lines on landscape patterns, pertinent policies should carefully consider complicated interactions at different spatial scales from a longer-term perspective, especially for especially ecological fragile areas, such as the Qinghai–Tibet Plateau [12,23,50].

## 5. Conclusions

The ecological impacts of the construction, operation, and development of the transportation infrastructure on the QTP cannot be overlooked. The aim of this study was to explore changes in the *LF* and *AEV* and their relationship before, during, and after the construction of the QTR so as to clarify the potential contributing influencing factors of the ecological impacts of the construction and operation of the QTR.

The results of both the *LF* and *AEV* showed that the sections, buffers, and bilateral sides had significant heterogeneous effects. There were clear distinctions among the 32 sections, the 22 buffers, and the bilateral study units. According to the results of the *LF*, the southern sections were relatively higher, although stability tended to be found in buffers larger than 2 km, with a more limited scope of influence than in previous studies. Additionally, the increase in landscape fragmentation was not restricted to the area around the railway. In terms of the *AEV* results, this was somewhat positively connected to the buffer width, and the difference between the two sides was also substantial.

Additionally, while comparing the construction and operation periods, both the *LF* and *AEV* demonstrated recoverability of the impacts. From 2010 to 2020, the *LF* of the majority of buffers on both sides steadily recovered. The majority of the study units’ *AEV* results declined from 2000 to 2010, but they also recovered over the period of operation, indicating that the ecosystem surrounding the railway could recover during its operation period.

Although there was a prominent negative correlation between the *AEV* and *LF* in 2020, the negative influence mechanism between them was not fully verified. The change trends of the *LF* and *AEV* were influenced differently by various human and natural circumstances. However, distance from main settlements, such as provincial capitals and prefecture-level regions, and areas with lower population densities could be favorable conditions for synchronous recovery of the *AEV* and *LF.*

According to the findings discussed above, the ecological impacts of the Qinghai–Tibet Railway may have been exaggerated in previous studies. However, the construction of transport infrastructure, regional development, and ecological environment protection must all be considered simultaneously in a location with such a vulnerable ecological environment.

## Figures and Tables

**Figure 1 ijerph-20-04154-f001:**
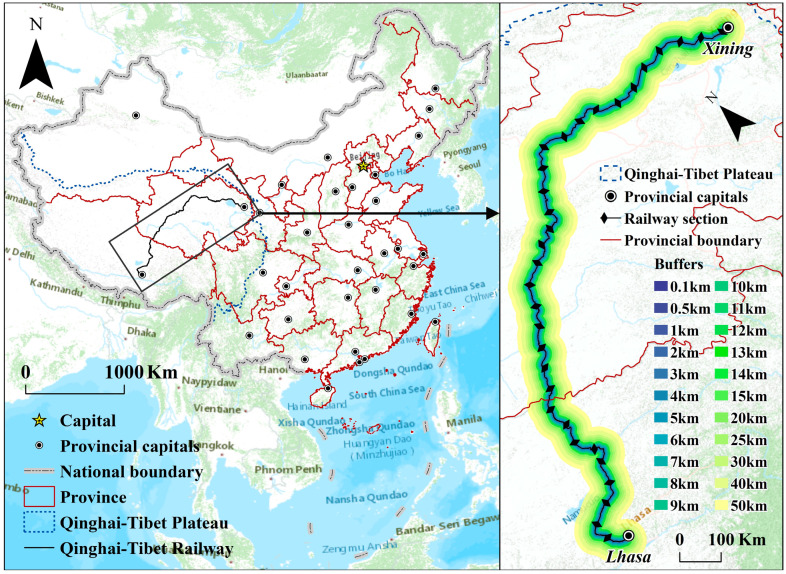
Location and different sections and buffers of QTR.

**Figure 2 ijerph-20-04154-f002:**
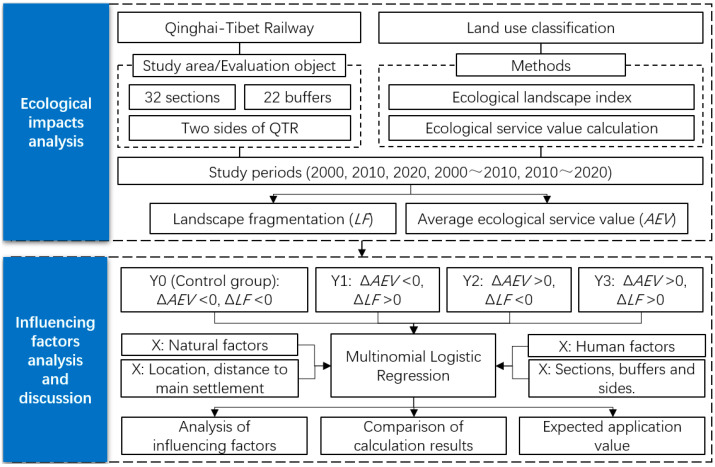
Analysis framework.

**Figure 3 ijerph-20-04154-f003:**
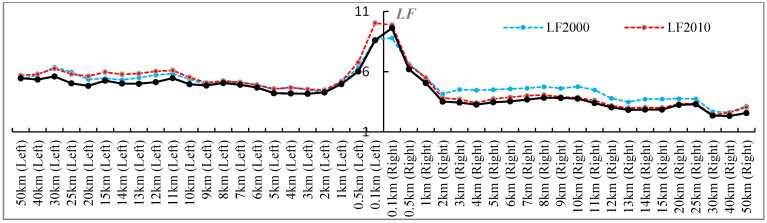
*LF* of different buffers on the left and right sides along the QTR.

**Figure 4 ijerph-20-04154-f004:**
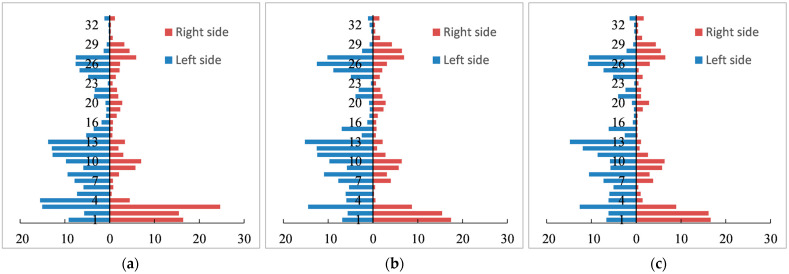
*LF* of 32 railway sections and different sides. (**a**) *LF* in 2000; (**b**) *LF* in 2010; (**c**) *LF* in 2020.

**Figure 5 ijerph-20-04154-f005:**
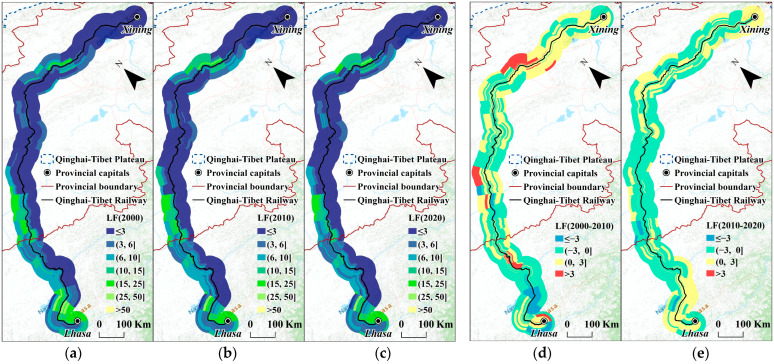
*LF* of different road sections and different buffers. (**a**) *LF* in 2000; (**b**) *LF* in 2010; (**c**) *LF* in 2020; (**d**) increase in *LF* from 2000 to 2010; (**e**) increase in *LF* from 2010 to 2020.

**Figure 6 ijerph-20-04154-f006:**
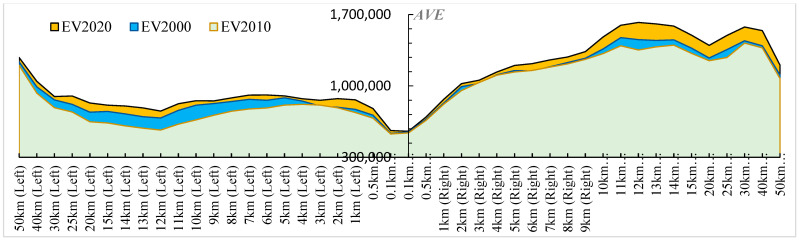
*AEV* of different buffers on the left and right sides along the QTR.

**Figure 7 ijerph-20-04154-f007:**
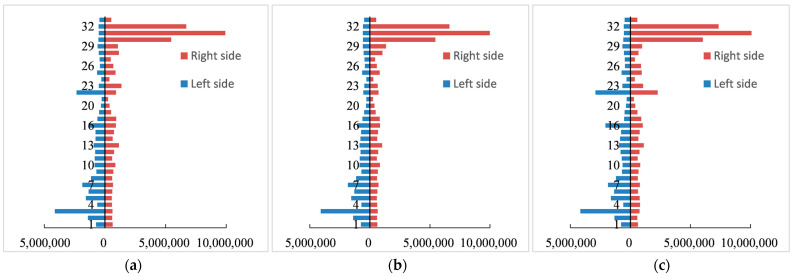
*AEV* of 32 different sections and different sides. (**a**) *AEV* in 2000; (**b**) *AEV* in 2010; (**c**) *AEV* in 2020.

**Figure 8 ijerph-20-04154-f008:**
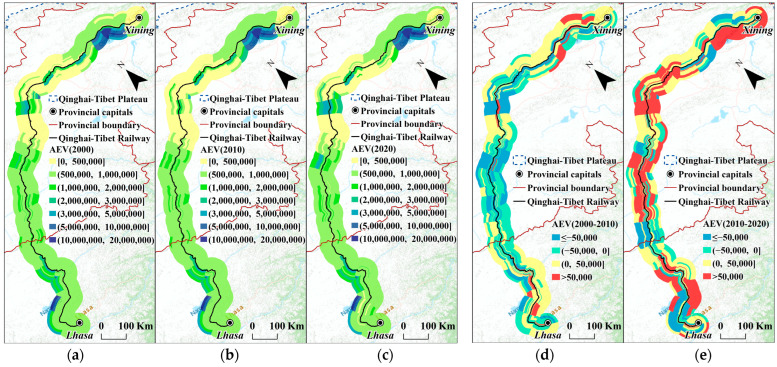
*AEV* of different road sections and different buffers. (**a**) *AEV* in 2000; (**b**) *AEV* in 2010; (**c**) *AEV* in 2020; (**d**) increase in *AEV* from 2000 to 2010; (**e**) increase in *AEV* from 2010 to 2020.

**Table 1 ijerph-20-04154-t001:** Ecological service value per unit area (*Va_i_*) of different land use types.

Land Use Classification	Ecological Service Value per Unit Area (*Va_i_*)
Farmland	4.01
Forest	17.53
Grassland	5.07
Shrubland	19.69
Wetland	52.02
Water body	125.61
Tundra, desert, and grassy marshland	6.265
Artificial surface	0
Bare land	0.2
Glaciers and perpetual snow	10.27

**Table 2 ijerph-20-04154-t002:** Selection and interpretation of variables.

Interpreted Variables	Explanatory Variables	
Y0: Increase in *AEV* < 0 while that of *LF* < 0 (Control group) Y1: Increase in *AEV* < 0 while that of *LF* > 0 (Worst-case scenario) Y2: Increase in *AEV* > 0 while that of *LF* > 0 (Relatively good case scenario) Y3: Increase in *AEV* > 0 while that of *LF* < 0 (Best-case scenario)	X_1_	DEM
	X_2_	Slope
	X_3_	Precipitation
	X_4_	Air temperature
	X_5_	In permafrost? True = 1, False = 0
	X_6_	Vulnerability of geological disasters
	X_7_	Distance to nearest provincial capital
	X_8_	Distance to nearest prefecture-level region
	X_9_	Distance to nearest county-level region
	X_10_	Population density
	X_11_	GDP density/nighttime light
	X_12_	Railway sections
	X_13_	Buffers
	X_14_	Which side of the railway? Left = 1, Right = 2

**Table 3 ijerph-20-04154-t003:** Pearson’s correlation between *LF* and *AEV*.

	*AEV* 2000	*AEV* 2010	*AEV* 2020	Increase in *AEV* from 2000 to 2010	Increase in *AEV* from 2010 to 2020
*LF* 2000	−0.047				
*LF* 2010		−0.038			
*LF* 2020			−0.062 *		
Increase in *LF* from 2000 to 2010				−0.015	
Increase in *LF* from 2000 to 2010					0.017

Note: * means it passed the 0.05 significance test.

**Table 4 ijerph-20-04154-t004:** Results of multinomial logistic regression analysis.

	2000–2010			2010–2020		
	Y1	Y2	Y3	Y1	Y2	Y3
x1_DEM	−0.502 *	0.754 *	−0.141	−1.808 **	−2.366 **	−1.328 **
x2_Slope	0.127	0.601 **	0.829 **	0.361 *	0.508 **	0.095
x3_ Precipitation	1.647 *	1.313	−2.305 **	−0.273	0.337	0.802 **
x4_Air temperature	0.247	0.167	0.355 *	−0.133	0.123	0.218
[x5_Not in permafrost]	−0.174	0.757 *	0.493	−0.303	0.334	−0.292
x6_ Vulnerability of geological disasters	0.057	−0.079	−0.158	−0.250	−0.573 **	−0.224
x7_ Distance to nearest provincial capital	−1.346 **	−0.556	0.820 *	−1.241 **	−0.071	0.635 *
x8_ Distance to nearest prefecture-level region	0.105	−0.312	−0.271	0.667 **	0.401 *	0.276 *
x9_ Distance to nearest county level region	−0.134	−1.153 **	−0.388 *	−0.047	0.332 *	0.298 *
x10_ Population density	0.042	0.021	−0.082	0.072	−0.139	−0.470 **
x11_ GDP density/nighttime light	0.105	0.314 *	−0.048	−0.547 **	−0.744 **	−0.072
x12_ Railway sections	−1.102	0.492	2.000 **	−1.143 **	−1.219 **	−0.562 *
x13_Buffer width	−0.056	0.039	0.101	−0.243	−0.044	0.143
x14_Which side of the QTR	−0.098	0.223	0.174	−0.287	−0.324	−0.274

Note: ** means the result passed the 0.01 significance test, while * means it passed the 0.05 significance test.

## Data Availability

All data used in this paper can be obtained publicly.

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
