# Peer review of "Ecological Impacts Associated with the Qinghai–Tibet Railway and Its Influencing Factors: A Comparison Study on Diversified Research Units"

_ijerph, 2023, doi:10.3390/ijerph20054154_

Round 1
Reviewer 1 Report
This paper attempts to analyze ecological changes along the Qinghai Tibet Railway, which is an important issue in an ecological fragile region. It is very interesting given its importance and the study area. However, there are still some problems to be modified.
1. Why 2000, 2010 and 2020 were chosen as the years for this study? Some explanation about the matching of different period of Qinghai-Tibet Railway and land use data should be further supplemented.
2. Although landscape fragmentation and ecological service value are the commonly used indicators in relevant research, the paper should further illustrate the reasons for using them. For example, what is the connotation of their difference? Why choose these indicators to judge the ecological changes?
3. The regression results of influencing factors need more in-depth analysis rather than listing them.
4. The discussions section can be revised and improved. I'd like to see more discussions on the policy implications with the comparison with the suggestions by previous researches in this area or other regions.
5. English need further grammar checking and polishing. It is also suggested to avoid long sentences.
Reviewer 2 Report
This paper explores the ecological impact of the Qinghai-Tibet Railway from the perspective of ecological fragmentation and ecological service value changes. The authors offer some new insights from the spatial structure perspective. However, the authors need to explain more clearly why they are addressing these two aspects. What are the implications of the synchronization of the two or not? How highlight the merit of the study is what the author needs to focus on.
In addition, the following suggestions are given for consideration.
1. The title is suggested to be simplified and focused.
2. The abstract needs to be added to the technical approach as well as the merits of the study.
3. The Globeland30 data name is the 2020 version, which is actually the 2017 data. Please note to check the data description.
4. what do Y1, Y2 and Y3 stand for in Table 4? The text does not seem to give a clear description.
5. The policy implications of the findings of the paper under discussion that can be directly produced need to be further refined.
6. It is suggested that the authors consider combining the conclusion and discussion sections or separating the two parts clearly.
Reviewer 3 Report
There are a few places that need to be supplemented or explained by the author.
1. The calculation formulas of △ LF and △ AEV are not given in the paper. Please add them and explain that if △ LF , LF2000-2010, and increment of LF from 2000 to 2010 are the same means.
2. Please add the data resource of the vector data of QTP.
3. Please supplement the processing methods of DEM, Slope and other spatial data.
4. What is the measurement unit of ecosystem service value?
5. In this paper the relationship between LF and AEV are carried out by Pearson’s correlation test. Please explain in the reply what the variables and the measurement method of variable relations involved in reference 42 or 43 is and whether it is consistent with that in this article.
There are many small problems such as spelling mistakes in the article. Please check the paper carefully. The following are examples of errors.
1. The distance unit and the reference of the chart need to be unified. For example, the distance unit in line 123 is “kilometer”, and that in line 138, 142 are “km”; the quote of figures 210 is “Fig. 3”, and that in line 223 is “Figure 4”.
2. “Table 1.” in 168 should be “Table 1”.
3. “Table 2” in 183 should be “Table 1”.
4. The Formula (2) is something wrong. And Should “Vi” in line 181 be “Vai”?
5. The method of relationship is “Pearson correlation” in line 297 but “Pearson’s correlation” in line 309.
